# Machine Learning Algorithms to Predict Mortality of Neonates on Mechanical Intubation for Respiratory Failure

**DOI:** 10.3390/biomedicines9101377

**Published:** 2021-10-02

**Authors:** Jen-Fu Hsu, Chi Yang, Chun-Yuan Lin, Shih-Ming Chu, Hsuan-Rong Huang, Ming-Chou Chiang, Hsiao-Chin Wang, Wei-Chao Liao, Rei-Huei Fu, Ming-Horng Tsai

**Affiliations:** 1Division of Pediatric Neonatology, Department of Pediatrics, Chang Gung Memorial Hospital, Taoyuan 33302, Taiwan; jeff0724@gmail.com (J.-F.H.); kz6479@cgmh.org.tw (S.-M.C.); qbonbon@gmail.com (H.-R.H.); cmc123@cgmh.org.tw (M.-C.C.); cyndi0805@cgmh.org.tw (H.-C.W.); rkenny@cgmh.org.tw (R.-H.F.); 2College of Medicine, Chang Gung University, Taoyuan 33302, Taiwan; cyulin@mail.cgu.edu.tw; 3Molecular Medicine Research Center, Chang Gung University, Taoyuan 33302, Taiwan; chiyang@mail.cgu.edu.tw (C.Y.); pettliao@mail.cgu.edu.tw (W.-C.L.); 4Department of Computer Science and Information Engineering, Asia University, Taichung 41354, Taiwan; 5Department of Nephrology, Chang Gung Memorial Hospital, Taoyuan 33305, Taiwan; 6Division of Neonatology and Pediatric Hematology/Oncology, Department of Pediatrics, Chang Gung Memorial Hospital, Yunlin 638, Taiwan

**Keywords:** neonatal mortality, artificial intelligence, big data analysis, early prediction, machine learning

## Abstract

Background: Early identification of critically ill neonates with poor outcomes can optimize therapeutic strategies. We aimed to examine whether machine learning (ML) methods can improve mortality prediction for neonatal intensive care unit (NICU) patients on intubation for respiratory failure. Methods: A total of 1734 neonates with respiratory failure were randomly divided into training (70%, *n* = 1214) and test (30%, *n* = 520) sets. The primary outcome was the probability of NICU mortality. The areas under the receiver operating characteristic curves (AUCs) of several ML algorithms were compared with those of the conventional neonatal illness severity scoring systems including the NTISS and SNAPPE-II. Results: For NICU mortality, the random forest (RF) model showed the highest AUC (0.939 (0.921–0.958)) for the prediction of neonates with respiratory failure, and the bagged classification and regression tree model demonstrated the next best results (0.915 (0.891–0.939)). The AUCs of both models were significantly better than the traditional NTISS (0.836 (0.800–0.871)) and SNAPPE-II scores (0.805 (0.766–0.843)). The superior performances were confirmed by higher accuracy and F1 score and better calibration, and the superior and net benefit was confirmed by decision curve analysis. In addition, Shapley additive explanation (SHAP) values were utilized to explain the RF prediction model. Conclusions: Machine learning algorithms increase the accuracy and predictive ability for mortality of neonates with respiratory failure compared with conventional neonatal illness severity scores. The RF model is suitable for clinical use in the NICU, and clinicians can gain insights and have better communication with families in advance.

## 1. Introduction

Despite innovations in perinatal resuscitation and advances in neonatal care, the in-hospital mortality rate for neonatal intensive care unit (NICU) patients has remained unchanged at 6.4–10.9% over the last decade [1,2,3,4]. NICU mortality is influenced by multiple factors, including underlying chronic comorbidities, artificial device-associated nosocomial infections, immature immune defense, and prolonged intubation [4,5,6]. Respiratory failure is one of the most important problems in the NICU, and 19.7–34% of total admissions have experienced respiratory failure [7,8,9]. In addition, respiratory failure is always the most common issue preceding the final mortality of preterm or critically ill neonates [10,11].

NICU scoring systems have been developed using a variety of admission factors to help prognosis prediction and communications between clinicians and parents [12,13,14]. However, it is often time-consuming to input the data, and these models often lack incorporation of important variables, including the influence of NICU characteristics, interventions, and therapeutic responses [12,15,16,17]. These limitations can be overcome by newly developed machine learning (ML) methods that make use of the increased computational capability to handle large amounts of linear and nonlinear parameters and time-series features [18,19]. Greater performance and excellent predictive power of the ML models can be achieved through deep learning and repeated validation [18,19,20]. An effective and alternative approach is required since previous neonatal scoring systems often fail to analyze numerous variables with nonlinearity and complex relationships in critically ill neonates. In this study, we aimed to develop and validate an ML algorithm that can accurately predict the in-hospital mortality of neonates with respiratory failure in the NICU.

## 2. Methods

### 2.1. Patients, Setting, and Study Design

A total of 1760 neonates who received intubation due to severe respiratory failure in the NICUs of Taipei and Linkou Chang Gung Memorial Hospital (CGMH) between January 2013 and December 2019 were retrospectively reviewed. The NICUs of Taipei and Linkou CGMH contain a total of four units and a total capacity of 57 beds equipped with ventilators and 70 beds of special care nurseries. The annual number of inpatients in these NICUs is 900 and accounts for approximately 30% of all critically ill and premature infants in Taiwan. 

Respiratory failure in the present study was defined when mechanical intubation was required to maintain a SpO_2_ value of 85–95%, CO_2_ 45–55 cmH_2_O, and pH 7.35–7.45 and/or the presence of hypotension that required cardiac inotropic agents and intravascular volume expansion. In our institute, all neonatologists follow the standard guideline that mechanical intubation will be done if we fail to maintain PaO_2_ > 60, a pH value > 7.25, and the requirement of fraction of inspired oxygenation (FiO_2_) > 60 using a noninvasive ventilator. Neonates who had severe congenital anomalies (*n* = 8), those with missing data on outcomes (*n* = 18), and those who died within the first day after intubation were excluded. Thus, 1734 neonates were analyzed in the present study. The subjects were randomly divided into a training set (70.0%, *n* = 1214) to develop the models and a test set (30.0%, *n* = 520) to test the performance of each model. This study was approved by the institutional review board of CGMH, with a waiver of informed consent because the waiver does not adversely affect the rights and welfare of the participants.

### 2.2. Study Variables

The onset of respiratory failure was defined when intubation was done and mechanical ventilation was used for the first time. For neonates successfully weaned from ventilators and reintubated during hospitalization in the NICU, only the first time of each patient was considered. Baseline patient demographics; the presence of artificial devices; and chronic comorbidities, including neurological sequelae, bronchopulmonary dysplasia (BPD) with/without pulmonary hypertension, symptomatic patent ductus arteriosus, cholestasis, renal function impairment, and gastrointestinal sequelae, were confirmed at the onset of respiratory failure. The laboratory data including white blood cell count, hemoglobin, platelet count, C-reactive protein, electrolytes, bilirubin, and renal and hepatic function results were measured at the timing of respiratory failure. 

In our institute, the initiation of mechanical intubation and shift to high-frequency oscillatory ventilation depend on the decisions of the attending physicians, but most clinicians follow the basic guidelines of the updated textbook of neonatology [21]. For ventilator settings and blood gas analyses, four time periods (at onset of respiratory failure (t_0_), 1–12 h (t_1_), 12–24 h (t_2_), and 24–48 h (t_3_) after intubation) were evaluated (Figure 1). The alveolar–arterial oxygen tension difference (AaDO2) and oxygenation index (OI) were also calculated during these four time periods. At the onset of respiratory failure (defined as from 30 minutes before intubation until 1 hour after intubation), the Neonatal Therapeutic Intervention Scoring System (NTISS) score and Score for Neonatal Acute Physiology Perinatal Extension II (SNAPPE-II) were calculated based on the calculation methods presented in the original studies [14,16]. The primary outcome was the NICU mortality, and the discontinuation of critical care due to family requests to transfer to other hospitals was censored. 

### 2.3. Statistical Analysis

Statistical analyses were performed using SPSS version 15.0 (SPSS, Chicago, IL, USA) software. Categorical and continuous variables were expressed as proportions and the median (interquartile, IQR), respectively. Categorical variables were compared by the χ^2^ test or Fisher’s exact test; odds ratios (ORs) and 95% confidence intervals (CIs) were calculated. Continuous variables were compared by the Mann–Whitney *U*-test and the t-test, depending on the distributions. R software (version 4.0.3) was used to construct mortality prediction models, and several machine learning algorithms were used, including artificial neural network (ANN), k-nearest neighbor (KNN), support vector machine (SVM), random forest (RF), and extreme gradient boost (XGB), bagged classification and regression tree (bagged CART), and elastic-net regularized logistic linear regression. The R package caret (version 6.0-86, https://github.com/topepo/caret) was used to train these predictive models with hyperparameter fine-tuning. For each of the ML algorithms, we performed 5-fold cross-validations of five repeats to determine the optimal hyperparameters that generate the least complex model within 1.5% of the best area under the receiver operating characteristic curve (AUC). The hyperparameter sets of these algorithms were predefined in the caret package, such as the mtry (number of variables used in each tree) in the RF model, the k (number of neighbors) in the KNN model, and the cost and sigma in the SVM model with the radial basis kernel function. The SVM models using kernels of linear, polynomial, and radial basis functions were constructed. We selected the radial kernel function for the final SVM model due to the highest AUC. Similar to SVM, the XGB model contains linear and tree learners. We applied the same highest AUC strategies and selected the tree learner for the final XGB model. When constructing each of the machine learning models, features were preselected based on the normalized feature importance to exclude irrelevancy. Then, the remaining features were considered to train the final models. 

Once the models were developed using the training set, the F1 score, accuracy, and areas under the curves (AUCs) were calculated on the test set to measure the performance of each model. For the predictive performance of the two traditional scores, NTISS and SNAPPE-II, we used Youden’s index as the optimal threshold of the receiver operating characteristic (ROC) curve to determine the probability of mortality, and the accuracy and F1 score were calculated. The AUCs of the models were compared using the DeLong test. We also assessed the net benefit of these models by decision curve analysis [22,23]. We converted the NTISS and SNAPPE-II scores into predicted probabilities with logistic regressions. We also assessed the agreement between predicted probabilities and observed frequencies of NICU mortality by calibration belts [24]. Finally, we used Shapley additive explanation (SHAP) values to examine the accurate contribution of each feature or input within the best prediction model [25]. All *P* values were two-sided, and a value of less than 0.05 was considered significant.

## 3. Results

In our cohort, 1214 (70.0%) neonates and 520 (30.0%) neonates with respiratory failure were randomly assigned to the training and test sets, respectively. The patient demographics, etiologies of respiratory failure, and most variables were comparable between these two sets (Table 1). In our cohort, more than half (55.9%) of our patients were extremely preterm neonates (gestational age (GA) < 28 weeks), and 56.5% were extremely low birth weight infants (BBW < 1,000g). Among neonates with respiratory failure requiring mechanical intubation, 83.1% of instances of respiratory failure occurred in the first week of life, and 65.1% occurred in the first day of life. A total of 278 (16.0%) patients died, and the in-hospital mortality rates were similar between the training and test sets. When the survivors were compared with those who finally died, many variables were different, including significantly higher severity of illness (higher NTISS and SNAPPE-II scores in the mortality group than among the survivors), lower birth weight and more preterm, more therapeutic interventions, and some disease entities (Appendix A).

### 3.1. Development of Mortality Prediction Model

Several ML models were developed using the training set and then validated in the test set. The F1 scores, accuracy, and AUC values resulting from the test set are presented in Table 2. The AUC value of the RF model was the highest among the ML models (0.939 (0.921–0.958)), and the next highest AUC value was achieved by the bagged CART (0.915 (0.891–0.939)). The RF models also achieved the highest accuracy and F1 score. The AUC values of the NTISS and SNAPPE-II for the prediction of in-hospital mortality were 0.836 (0.800–0.871) and 0.805 (0.766–0.843), respectively. Significant performance was achieved by both the RF and bagged CART models compared with the conventional scoring systems (NTISS and SNAPPE-II scores) (Figure 2A). The net benefit of both the RF and bagged CART models ranged from 3 to 100%, which significantly outperformed the ranges corresponding to the NTISS and SNAPPE-II scores (Figure 2b, without 95% confidence intervals (CIs); Appendix A, with 95% CI).

Among the machine learning models, the performances of the RF, bagged CART, and SVM models were significantly better than those of the XGB, ANN, and KNN models (Appendix A). The RF and bagged CART models also had significantly higher accuracy and F1 scores than the XGB, ANN, and KNN models. In addition, the RF model has a significantly better AUC value than the bagged CART model. 

The calibration belts of the RF and bagged CART models and the conventional scoring systems for NICU mortality prediction are shown in Figure 3. The RF model showed better calibration among neonates with respiratory failure who were at a high risk of mortality than did the NTISS and SNAPPE-II scores, especially when the predicted values were higher than 0.8–0.83.

### 3.2. Rank of Predictors in the Prediction Model

A total of 41 variables or features were used to develop the prediction model. Of these variables, 18 (43.9%) were indicative of therapeutic response at the t1, t2, and t3 time periods, and only 5 (12.2%) indicated the initial severity of illness. Although certain disease entities were significantly associated with a higher risk of final in-hospital mortality (Appendix A), none of them was in the final RF prediction model. The importance matrix plot for the RF method is shown in Figure 4, which reveals that the top five most important variables contributing to the model were the OI value at t3, the AaDO_2_ values at t3, the PH value at the onset of respiratory failure, the OI value at t2, and the initial PaO_2_. 

We depicted the SHAP summary plot of RF using the top 20 features of the prediction model to identify the most important features that influenced the prediction model (Figure 5). A feature with a higher SHAP value indicates a higher likelihood of NICU mortality based on the prediction model. The red and blue plots in the SHAP represent larger and smaller values, respectively, which suggest that increasing values or decreasing values will increase or decrease the predicted probability of mortality, respectively. The SHAP is consistent with the perfect performance of our RF model.

## 4. Discussion

In the NICU, respiratory failure and the need for mechanical intubation often indicate a higher severity of illness and that the patient is at risk of death. We developed an RF model trained on 41 binary and continuous variables from more than 1200 neonates hospitalized in four tertiary-level NICUs of medical centers in Taiwan. We found that the RF and bagged CART models have significantly better predictive ability than the traditional neonatal severity scoring systems including the NTISS and SNAPPE-II. The clinically applicable RF model was explainable, the top important features were identified, and this model was confirmed to be superior to other ML methods using calibration, decision curve analyses, and SHAP methods.

Using machine learning algorithms to help clinicians has formed a major emerging research trend in the past decade [18,19,20,24,25,26,27]. The mortality of critically ill neonates with respiratory failure has previously been difficult to predict because most neonates can survive the initial critical period and various life-threatening events may occur during their long-term hospital courses [28]. Therefore, the successful development of an ML model to accurately predict the final outcomes of neonates with respiratory failure, most instances of which occurred in the first week of life, is very important for clinicians’ insights and early communication with families. In addition, although some disease entities were associated with a significantly higher risk of in-hospital mortality, none of them were in the final RF model. We found that nearly half of the top 20 features or variables on the importance matrix plot and the SHAP summary plot of RF were parameters of therapeutic responses, which demonstrated the value of data on the first and second days of respiratory failure and highlighted the importance of the initial therapeutic strategies.

Various neonatal scoring systems for illness severity have been applied to predict outcomes of NICU patients, including SNAPPE-II, NTISS, Score for Neonatal Acute Physiology II (SNAP II), and Modified Sick Neonatal Score (MSNS) [13,14,16]. Most of the scoring systems have the advantages of high applicability, easy interpretation, and acceptable predictive power (an AUC of approximately 0.86–0.91 for the prediction of mortality) [16,29,30]. However, the discriminative abilities of these scores will be influenced by different cutoff points and the therapeutic interventions of different clinicians [16,31,32], which limit their clinical applications in decision-making, especially at the most critical time point [13,14]. Therefore, an AUC value of 0.80–0.83 was found in our cohort, which is relatively lower [31,32,33], because most of the neonates in our cohort had higher illness severity. Mesquitz et al. recently concluded that the discriminative abilities of SNAP II and SNAPPE-II scores to predict in-hospital mortality were only moderate [34]. Instead, a machine learning model incorporating parameters of therapeutic responses may be more suitable for clinicians’ judgments, because we found that the important predictive features were actionable or could be manipulated by the decisions of clinicians. Because many parameters of therapeutic responses were in the final RF model, it is necessary to build a statistical and causal model that investigates how physiological factors interact with and react to interventions. Therefore, the next step to make this model clinically applicable will be randomized clinical trials.

Among the various machine learning models, we found that decision tree-based methods, including RF and bagged CART, had superior performances compared to nonlinear methods of ANN or KNN. This observation is also consistent with other ML models recently developed for medical use [24,35]. Although the tree learner method was applied in the XGB method, the performance of XGB was the worst in this study. Therefore, we can conclude that the bootstrap aggregating method of RF and bagged CART was more suitable than the boosting method of XGB to improve the stability, increase accuracy, reduce variance, and help to avoid overfitting [36]. 

The decision curve analysis is used to identify the net benefit of performing various different ML models at different risk levels and assessing the utility of models for decision-making [20,21]. The model with a high decision curve analysis can help clinicians in screening patients who are at higher risk of final mortality. In our analysis, both the RF and bagged CART models improved the net benefit for predicting the NICU mortality than the traditional severity scores at a very wide range of threshold probabilities. Therefore, we showed the threshold range above the prediction curve in the analysis, which indicates the applicability of our ML algorithms in clinical practice.

In addition, we also applied SHAP to calculate the contribution of each feature to the RF model. The concept of SHAP is to average the difference between the predicted values with and without the effect of adding each feature for all combinations and examine the influence of inputs after machine learning [25,37]. SHAP was used in this study because most ensemble ML methods have the disadvantage of decreased interpretability after achieving high accuracy. Because ensemble learning methods are usually more complex than traditional learning methods, SHAP can be applied to understand the importance of each feature and which direction the feature affects [25,37]. Thus, we can build a highly predictive model with good transparency for the outcomes.

There are some limitations in the current study. First, the generalizability of this model to other institutes has not been determined since this is only a two-center study and this model has not been validated in a prospective setting. Second, it is inevitable that some important variables in the training set had missing data in the retrospective study design of the present study. Although the RF model is proven highly predictive, a great number of features and therapeutic responses need to be incorporated and complex computer analyses are needed, which may limit its applicability. Another limitation is the inability of SHAP values to resolve algorithmic bias [38], because most ML models do not have underlying causal structure and make predictions based only on the majority of cases [39]. Last, our RF model incorporated many features of therapeutic responses, which were mostly obtained at 12 to 24 h or the second day of respiratory failure; therefore, this model cannot provide real-time prediction of mortality at the onset of respiratory failure. 

## 5. Conclusions

It has been difficult to estimate the probability of mortality for neonates with respiratory failure. We demonstrated that the RF and bagged CART models can more accurately predict the mortality of critically ill neonates than conventional scoring systems such as the NTISS and SNAPPE-II scores. These results highlight the applicability of using ML algorithms for clinical use in the NICU. Further studies are indicated to examine whether machine learning can also help clinicians make more prompt decisions.

## Figures and Tables

**Figure 1 biomedicines-09-01377-f001:**
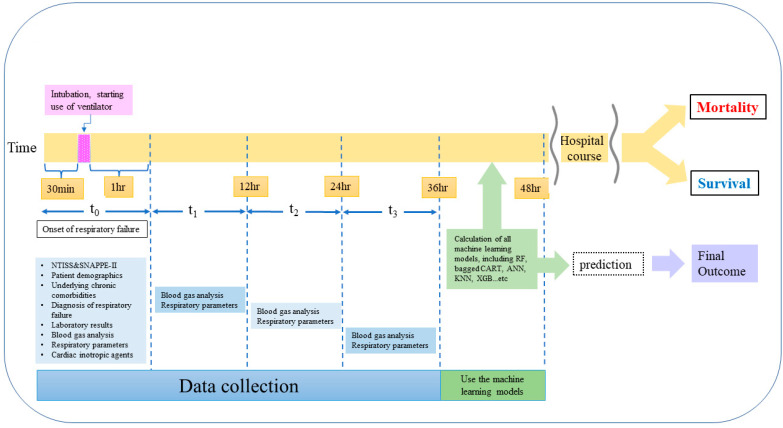
Time period and time point to collect the whole features and variables of the training and test sets. The study design highlights that the clinically applicable machine model can be used on the second day of respiratory failure to predict the in-hospital mortality of neonates with respiratory failure.

**Figure 2 biomedicines-09-01377-f002:**
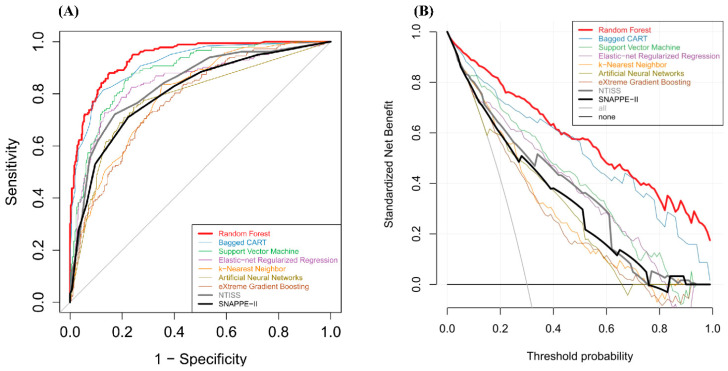
Comparisons of neonatal intensive care unit mortality prediction models such as random forest, NTISS, and SNAPPE-II in the test set. (**A**) Receiver operating characteristic curves of all machine learning models, the NTISS, and the SNAPPE-II. (**B**) Decision curve analysis of all machine learning models, the NTISS, and the SNAPPE-II. Bagged CART: bagged classification and regression tree; NTISS: Neonatal Therapeutic Intervention Scoring System; SNAPPE-II: Score for Neonatal Acute Physiology Perinatal Extension II.

**Figure 3 biomedicines-09-01377-f003:**
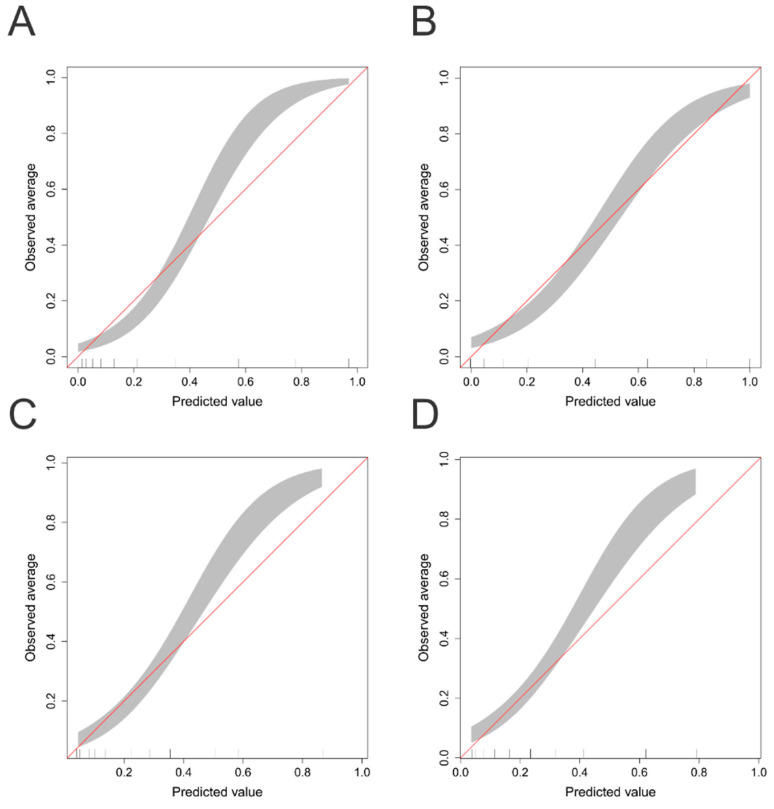
Calibration belts of (**A**) random forest, (**B**) bagged classification and regression tree (bagged CART), (**C**) NTISS, and (**D**) SNAPPE-II for NICU mortality prediction in the test set.

**Figure 4 biomedicines-09-01377-f004:**
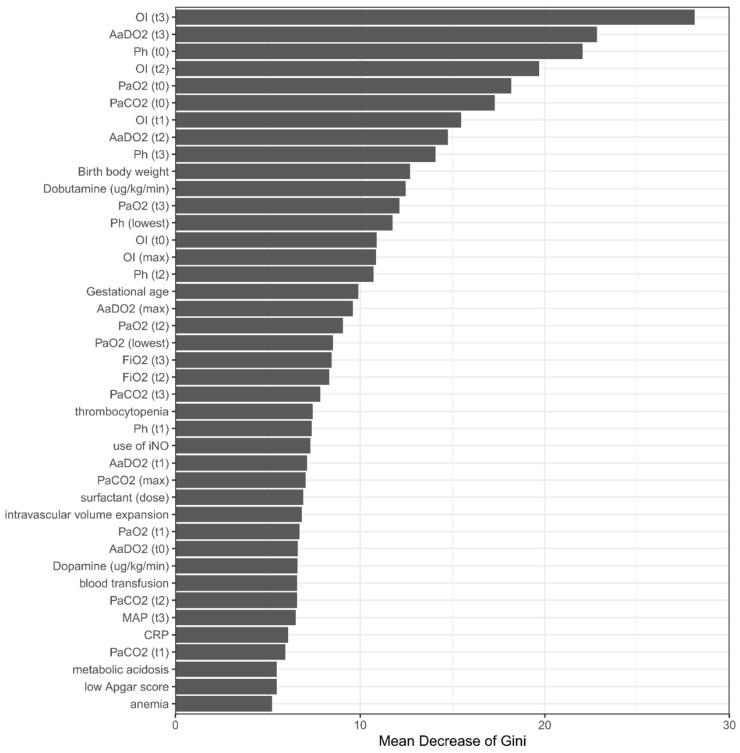
Importance matrix plot of the RF model. This importance matrix plot depicts the importance of each covariate in the development of the final predictive model. Abbreviations: OI: oxygenation index; AaDO2: alveolar–arterial oxygen tension difference; MAP: mean airway pressure; FiO2: fraction of inspired oxygen.

**Figure 5 biomedicines-09-01377-f005:**
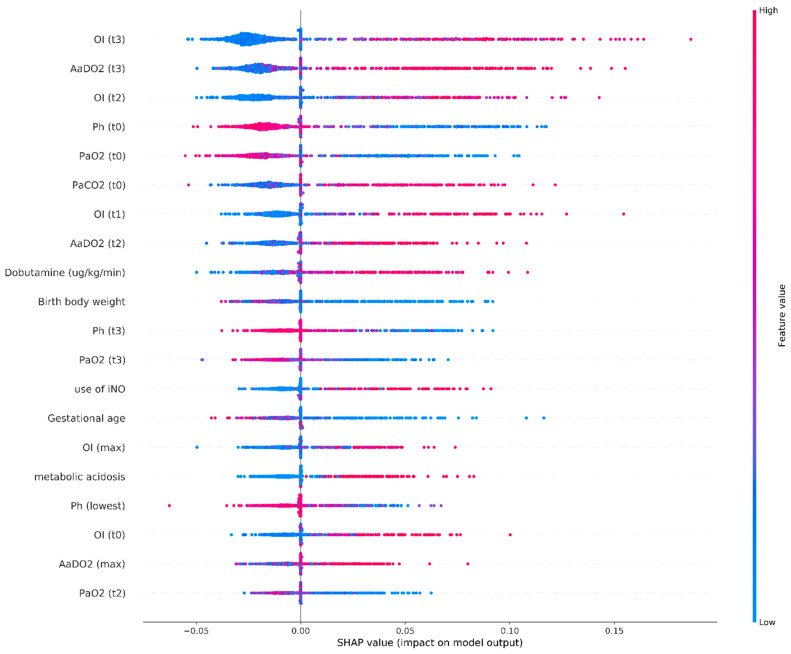
SHAP summary plot of the top 20 features of the RF model. The higher the SHAP value of a feature, the higher the probability of mortality in NICU patients with respiratory failure. Each dot is made up of each feature attribution value to the model of each patient. Red dots and blue dots represent higher feature values and lower feature values, respectively. Abbreviations: OI: oxygenation index; AaDO2: alveolar–arterial oxygen tension difference.

**Table 1 biomedicines-09-01377-t001:** Patient demographics, characteristics, and clinical presentation of all neonates with respiratory failure.

Characteristics	All Study Subjects (Total *n* = 1734)	The Training Set (Total *n* = 1214)	The Test Set (Total *n* = 520)	*p* Values
Case demographics				
Gestational age (weeks), median (IQR)	27.0 (25.0–31.3)	27.3 (25.3–31.0)	27.0 (25.0–31.5)	0.324
Birth weight (g), median (IQR)	915.0 (703.5–1480.0)	915.0 (708.0–1463.8)	908.5 (700.0–1510.0)	0.974
Gender (male), *n* (%)	1029 (59.3)	732 (60.3)	297 (57.1)	0.220
Birth by NSD/Cesarean section, *n* (%)	548 (31.6)/1186 (68.4)	389 (32.0)/825 (68.0)	159 (30.6)/361 (69.4)	0.573
5 minutes Apgar score < 7, *n* (%)	566 (32.6)	409 (33.7)	157 (30.2)	0.163
Inborn/outborn, *n* (%)	1365 (78.7)/369 (21.3)	944 (77.8)/270 (22.2)	421 (81.0)/99 (19.0)	0.141
Premature rupture of membrane, *n* (%)	530 (30.6)	375 (30.9)	155 (29.8)	0.691
Maternal fever, *n* (%)	214 (12.3)	156 (12.9)	58 (11.2)	0.340
Intrapartum antibiotic prophylaxis, *n* (%)	140 (8.1)	90 (7.4)	50 (9.6)	0.125
Chorioamnionitis, *n* (%)	32 (1.8)	22 (1.8)	10 (1.9)	0.848
Perinatal asphyxia, *n* (%)	354 (20.4)	255 (21.0)	99 (19.0)	0.363
Onset of respiratory failure, day (median (IQR))	1.0 (1.0–3.0)	1.0 (1.0–3.0)	1.0 (1.0–2.0)	0.101
Diagnoses of respiratory failure, *n* (%)				
Respiratory distress syndrome (≥Gr II)	1047 (60.3)	736 (60.5)	311 (59.8)	0.748
Transient tachypnea of newborn	83 (4.8)	61 (5.0)	22 (4.2)	0.540
Complicated cardiovascular diseases	28 (1.6)	22 (1.8)	6 (1.2)	0.408
Symptomatic patent ductus arteriosus	662 (38.2)	454 (37.4)	208 (40.0)	0.306
Persistent pulmonary hypertension of newborn	278 (16.0)	187 (15.4)	91 (17.5)	0.284
Pulmonary hemorrhage	120 (6.9)	91 (7.5)	29 (5.6)	0.179
Pneumonia	85 (4.9)	54 (4.4)	31 (6.0)	0.183
Air leak syndrome ^&^	188 (10.8)	124 (10.2)	64 (12.3)	0.206
Meconium aspiration syndrome	50 (2.9)	35 (2.9)	15 (2.9)	1.000
Sepsis	271 (15.6)	190 (15.7)	81 (15.6)	1.000
Hydrops fetalis	34 (2.0)	27 (2.2)	7 (1.3)	0.261
Others ^#^	26 (1.5)	20 (1.6)	6 (1.2)	0.523
Presences of any chronic comorbidities, *n* (%)	379 (21.8)	272 (22.4)	107 (20.6)	0.410
Presences of central venous catheter, *n* (%)	522 (30.1)	351 (28.9)	171 (32.9)	0.109
Initial ventilator requirement *, *n* (%)				0.178
Intubation with mechanical ventilation	1168 (67.4)	819 (67.5)	349 (67.1)	
Initial FiO_2_ ≤ 50	671 (38.7)	464 (38.2)	207 (39.8)	
Initial FiO_2_ > 50	497 (28.7)	355 (29.2)	142 (27.3)	
On high frequency oscillatory ventilation	566 (32.6)	395 (32.5)	171 (32.9)	
High setting (FiO_2_ ≤ 50)	248 (14.3)	162 (13.3)	86 (16.5)	
Low setting (FiO_2_ > 50)	318 (18.3)	233 (19.2)	85 (16.3)	
Oxygenation index, median (IQR)	11.0 (6.0–20.0)	11.0 (6.0–20.0)	11.0 (6.0–20.0)	0.780
AaDO_2_, median (IQR)	273.0 (166.0–478.0)	271.0 (165.8–477.0)	280.0 (166.3–486.8)	0.670
Use of iNO	285 (16.4)	193 (15.9)	92 (17.7)	0.359
Clinical features *, *n* (%)				
Intravascular volume expansion	1415 (81.6)	1001 (82.5)	414 (79.6)	0.188
Requirement of cardiac inotropic agents	1206 (69.6)	842 (69.4)	364 (70.0)	0.820
Metabolic acidosis	677 (39.0)	484 (39.9)	193 (37.1)	0.307
Coagulopathy	1226 (70.7)	864 (71.2)	362 (69.6)	0.527
Requirement of blood transfusion **	558 (32.2)	391 (32.2)	167 (32.1)	1.000
Laboratory data at onset of respiratory failure				
Leukocytosis or leukopenia	446 (25.7)	304 (25.0)	142 (27.3)	0.337
Shift to left in WBC (immature > 20%)	158 (9.1)	103 (8.5)	55 (10.6)	0.172
Anemia (hemoglobin level < 11.5 g/dL)	317 (18.3)	223 (18.4)	94 (18.1)	0.946
Thrombocytopenia (platelet < 150,000/ul)	434 (25.0)	297 (24.5)	137 (26.3)	0.432
C-reactive protein (mg/dL), median (IQR)	5.0 (2.0–19.5)	5.0 (2.0–18.0)	6.0 (2.0–22.0)	0.149
Severity score at onset of respiratory failure				
NTISS (median (IQR))	23.0 (21.0–26.0)	24.0 (21.0–27.0)	23.0 (21.0–26.0)	0.254
SNAPPE-II (median (IQR))	28.0 (22.0–40.0)	30.0 (22.0–42.0)	28.0 (22.0–40.0)	0.174
Final in-hospital mortality, *n* (%)	278 (16.0)	198 (16.3)	80 (15.4)	0.379

FiO_2_: fraction of inspired oxygen; NSD: normal spontaneous delivery; IQR: interquartile range; iNO: inhaled nitric oxide; HFOV: high-frequency oscillatory ventilator; WBC: white blood cell; NTISS score: Neonatal Therapeutic Intervention Scoring System; SNAPPE-II: Score for Neonatal Acute Physiology Perinatal Extension II; ^&^ including pneumothorax, pneumomediastinum, and pulmonary interstitial emphysema; * at onset of respiratory failure; ** including leukocyte-poor red blood cell and/or platelet transfusion; ^#^ including congenital diaphragmatic hernia (21), pulmonary sequestration (3), and pulmonary dysplasia (2).

**Table 2 biomedicines-09-01377-t002:** Mortality prediction models for neonates on mechanical ventilation for severe respiratory failure in the test set.

Models	AUC (95% CI)	*p* Value *	*p* Value ^#^	Accuracy	F1 Score
NTISS	0.836 (0.800–0.871)			0.701	0.629
SNAPPE-II	0.805 (0.766–0.843)			0.757	0.637
Random Forest	0.939 (0.921–0.958)	< 0.0001	< 0.0001	0.877	0.777
Bagged CART	0.915 (0.891–0.939)	0.0003	< 0.0001	0.864	0.774
Support Vector Machine	0.884 (0.856–0.912)	0.0343	0.0010	0.833	0.720
Elastic-net Regularized Regression	0.844 (0.888–0.934)	0.7409	0.1386	0.849	0.754
k-Nearest Neighbor	0.795 (0.759–0.832)	0.1200	0.7345	0.698	0.613
Artificial Neural Networks	0.782 (0.742–0.822)	0.0487	0.4186	0.773	0.635
eXtreme Gradient Boosting	0.776 (0.737–0.815)	0.0307	0.2981	0.719	0.582

NTISS: Neonatal Therapeutic Intervention Scoring System; SNAPPE-II: Score for Neonatal Acute Physiology Perinatal Extension II; AUC: area under the curve, 95% CI: 95% confidence interval; * compared with NTISS score; ^#^ compared with SNAPPE-II score.

## Data Availability

The datasets used/or analyzed during the current study are available from the corresponding author on reasonable request.

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
