# Peer review of "Machine Learning Algorithms to Predict Mortality of Neonates on Mechanical Intubation for Respiratory Failure"

_biomedicines, 2021, doi:10.3390/biomedicines9101377_

Round 1

Reviewer 1 Report

This is a well written paper comparing the ability of a range of machine learning algorithms to predict mortality of neonates who are on mechanical intubation for respiratory failure. The study found that Random Forest provided the highest predictability and out-performed the traditional NTISS and SNAPP-II scores.

The authors provide a well-structured approach using demographic and early respiratory support, supplement oxygen and illness severity measures with increased OI at day of life 3 being the largest contributor to the mortality prediction model.

Comments:

  1. Fig 4- It is interesting to note that OI (t3) is the biggest contributor to the model yet the components of this index (FiO2, MAP and PaO2) where much lower on the importance list. How does the RF handle these closely related variables? On a similar note OI(t0) was an important covariate as well as PaO2(t0) but FiO2(t0) and MAP(t0) aren’t on the list. Since OI=(FiO2xMAP)/PaO2 does that mean that the contribution of OI(t0) was all due to PaO2(0)?

Minor comments-

  1. initial use of abbreviations should be spelled out
  2. Table 1 and S1. Some data entries can be simplified. For example, Gender (male), n(%) 1029 (59.3)
  3. Figure legend for Supplemental Fig 1 would be very helpful

Author Response

Dear Reviewer:

Best regard,

Tsai Ming Horng

Reviewer 2 Report

This article purports to be a study of neonates with respiratory failure, but the patient demographics suggest the vast majority of these babies were premature.  Can the authors please clarify their target population?  This is important as predictive analytics might differ between premature versus term infants who would suffer respiratory failure from a variety of other pathologies. Furthermore, some of the comparison models were limited to preterm populations.

It is suggested that the authors expand their model analysis to include the PhysiScore ("Integration of early physologic responses predicts later illness everity in premature infants" Saria S., et al. Sci Transl Med 2010; 2:48ra65) which uses only 10 patient characteristics at 3 hours of age to predict mortality in premature infants. They should also include in their discussion and references a recent article by Dr. F. Sessions Cole ("Improving VLBW infant outcomes with big data analytics" Pediatr Res. 2021 Jul;90(1):20-21.)

As pointed out in their Discussion, "our RF model incorporated many features of therapeutic responses, which were mostly obtained at 12 to 24 hours or the 2nd day of respiratory failure; therefore, this model cannot provide real-time prediction of mortality at the onset of respiratory failure."  Can the authors compare this aspect of their approach to other predictive models?  How do they suggest their findings be used in any practical setting?  How could the improvement in predictability by random forest have any impact in clinical care? In their discussion, the authors state: "Therefore, the next step to make this model clinically applicable will be randomized clinical trials." What do they suggest the aim and methodologies of such RCTs be?

I am not sure if I understand how the Oxygenation Index for the whole dataset, the training set and the test set could all be identical in Table 1, with a P value of 0.78, when I think it would be 1.0.  Is there a typo in one or more of these columns?

The authors state: "discontinuation of critical care due to family requests to transfer to other hospitals was censored." How many patients does this include?  Is it possible that too many such patient transfers would compromise the findings of this study.

Page 3, Line 128 - please either explain or omit "etc"

Author Response

(The authors gave the same response as above.)

Reviewer 3 Report

This is a very large study (1760) neonates investigating whether machine learning (ML) methods can improve mortality prediction for neonatal intensive care unit (NICU) patients on intubation for respiratory failure compared to the conventional neonatal illness severity scoring systems including the NTISS and SNAPPE-II. The authors conclude that Machine learning algorithms increase the accuracy and predictive ability for mortality of neonates with respiratory failure compared with conventional neonatal illness severity scores. The RF model is suitable for clinical use in the NICU and clinicians can gain insights and have better communication with families in advance.

Among the machine learning models, the performances of the RF, bagged CART, and SVM models were significantly better than the XGB, ANN, and KNN models. In addition, the RF model has a significantly better AUC value than the bagged CART mode.

Most of the neonatal scoring systems have the advantages of high applicability, easy interpretation and acceptable predictive power and they are very quick to perform using few parameters. On the contrary RF model incorporated a great number of features of therapeutic responses, which were mostly obtained at 12 to 24 hours or the 2nd day of respiratory failure. Ι suggest that the authors include this in the limitations section, as well as the need of complex computer analysis.

This is a very interesting study and I suggest that although quite complex it is suitable for publication in the journal.

Author Response

(The authors gave the same response as above.)
